# Calcium Improves Germination and Growth of *Sorghum bicolor* Seedlings under Salt Stress

**DOI:** 10.3390/plants9060730

**Published:** 2020-06-10

**Authors:** Takalani Mulaudzi, Kaylin Hendricks, Thembeka Mabiya, Mpho Muthevhuli, Rachel Fanelwa Ajayi, Noluthando Mayedwa, Christoph Gehring, Emmanuel Iwuoha

**Affiliations:** 1Life Sciences Building, Department of Biotechnology, University of the Western Cape, Private Bag X17, Bellville 7535, South Africa; 3338080@myuwc.ac.za (K.H.); 2340865@myuwc.ac.za (T.M.); 3677780@myuwc.ac.za (M.M.); 2SensorLab, Department of Chemical Sciences, University of the Western Cape, Private Bag X17, Bellville 7535, South Africa; fngece@uwc.ac.za (R.F.A.); eiwuoha@uwc.ac.za (E.I.); 3iThemba Laboratory for Accelerator Based Science, Material Research Department, P.O. Box 722, Somerset West 7129, South Africa; nmyedi@gmail.com; 4Department of Chemistry, Biology & Biotechnology, University of Perugia, Borgo XX Giugno 74, 06121 Perugia, Italy; christophandreas.gehring@unipg.it

**Keywords:** antioxidant, Ca^2+^, gene expression, germination, ion homeostasis, NaCl, salt stress, oxidative stress, *Sorghum bicolor*

## Abstract

Salinity is a major constraint limiting plant growth and productivity worldwide. Thus, understanding the mechanism underlying plant stress response is of importance to developing new approaches that will increase salt tolerance in crops. This study reports the effects of salt stress on *Sorghum bicolor* during germination and the role of calcium (Ca^2+^) to ameliorate some of the effects of salt. To this end, sorghum seeds were germinated in the presence and absence of different NaCl (200 and 300 mM) and Ca^2+^ (5, 15, or 35 mM) concentrations. Salt stress delayed germination, reduced growth, increased proline, and hydrogen peroxide (H_2_O_2_) contents. Salt also induced the expression of key antioxidant (*ascorbate peroxidase and catalase*) and the *Salt Overlay Sensitive1* genes, whereas in the presence of Ca^2+^ their expression was reduced except for the *vacuolar Na^+^/H^+^ exchanger antiporter2* gene, which increased by 65-fold compared to the control. Ca^2+^ reversed the salt-induced delayed germination and promoted seedling growth, which was concomitant with reduced H_2_O_2_ and Na^+^/K^+^ ratio, indicating a protective effect. Ca^2+^ also effectively protected the sorghum epidermis and xylem layers from severe damage caused by salt stress. Taken together, our findings suggest that sorghum on its own responds to high salt stress through modulation of osmoprotectants and regulation of stress-responsive genes. Finally, 5 mM exogenously applied Ca^2+^ was most effective in enhancing salt stress tolerance by counteracting oxidative stress and improving Na^+^/K^+^ ratio, which in turn improved germination efficiency and root growth in seedlings stressed by high NaCl.

## 1. Introduction

Plants are affected by environmental factors throughout their life cycle and seed germination is the most critical stage that determines plant growth and productivity [1]. Seed germination is a complex physiological process, which begins when water is taken up by the seed and is completed by the appearance of the radicle [2]. Among other abiotic factors, such as high temperatures, heavy metals, and drought, salinity is one of the detrimental abiotic factors, causing up to 50% crop loss worldwide. About 20% of the cultivated land and 33% of all irrigated land worldwide are affected by salinization and this is expected to continue to worsen [3,4]. Salinity slows seed germination and hinders seedling growth, which could be a result of high osmotic pressure making it difficult for the seed to absorb water or by the effects of ions that are toxic to the embryo [5]. Osmotic stress is considered as the initial harmful effect of salinity on plants and is caused by decreased water absorption capacity and increased water loss from leaves. Ionic stress result from the accumulation of toxic Na^+^ and Cl^−^ ions, which cause ionic imbalances and prevent the uptake of essential ions such as potassium (K^+^) in plant tissues [6]. High NaCl concentration, causes Na^+^ to displace Ca^2+^ from membranes, increasing intracellular Na^+^, resulting in higher Na^+^/K^+^ ratio [7,8]. These events lead to nutrient imbalances, loss of membrane function, decreased photosynthesis, affects stomatal closure, and alters the ability of antioxidant enzymes to detoxify reactive oxygen species (ROS) [9]. ROS are chemical species that contain oxygen and commonly include; superoxide radical (O_2_^−^), hydrogen peroxide (H_2_O_2_), singlet oxygen (O_2_), and hydroxyl radical (OH^−^). ROS are highly toxic to plants when produced at high levels causing oxidative membrane damage, peroxidation of lipids, nucleic acids and proteins, ultimately leading to cell death [10,11]. ROS levels are regulated by both the non-enzymatic and enzymatic antioxidant systems. Enzymatic antioxidants include a set of enzymes namely, superoxide dismutase (SOD), peroxidase (POD), catalase (CAT) and enzymes of the ascorbate (ASC)-glutathione (GSH) cycle [ascorbate peroxidase (APX), glutathione reductase (GR), monodehydroascorbate dehydrogenase (MDHAR), and dehydroascorbate reductase (DHAR)] [11,12].

Compatible solutes, such as proline and carbohydrates contribute to the maintenance of the osmotic balance, stabilization of proteins and enzymes, and enable water absorption [13]. Ion transporters and channels such as the salt overly sensitive 1 (SOS1) and the vacuolar Na^+^/H^+^ exchanger (NHX) antiporter are some of the key components involved in ion homeostasis. Maintenance of ion homeostasis is also crucial to improve crop tolerance under salt stress. SOS1 is a plasma membrane Na^+^/H^+^ antiporter that enables Na^+^ efflux at the root surface and regulates its transport from the root to the shoot thereby maintaining the equilibrium of K^+^ and Ca^2+^ [14,15,16]. Na^+^ in the cytoplasm is transported into the vacuole for osmotic adjustment by NHX [17]. Ca^2+^ is a macronutrient that is beneficial in maintaining plant growth, development, and crop yield [18]. Ca^2+^ is not just a signaling molecule, but also a structural component of the cell wall and plasma membrane. It contributes to the strength of plant stems, improves the absorption of nutrients across the plasma membrane, influences the transfer of sugars, and helps form and stabilize organelles such as the mitochondria and nucleus [19]. In addition, Ca^2+^ functions as a secondary messenger that mediates the metabolic processes of plant growth, development, and stress tolerance [18,20,21]. It also alleviates ionic stress by binding to the plasma membrane and blocks the non-selective cation channels that are the major pathways for Na^+^ influx in plants [22,23]. Several studies demonstrated the effectiveness of exogenous Ca^2+^ to improve seed germination through maintaining ion homeostasis and modulating the antioxidant system under salt stress [8,24,25]

Thus, this study was conducted to elucidate aspects of salt stress tolerance mechanisms in *Sorghum bicolor* and the ability of exogenous Ca^2+^ to improve tolerance through enhancing the antioxidant and ion homeostasis systems. Sorghum is a C4 plant that is particularly adapted to grow and yield in arid and semi-arid areas and is moderately tolerant of drought and salt [26,27]. Hence, it is possible to take advantage of these characteristics and study the mechanisms of salt tolerance in order to develop ways to mitigate salt stress in cereals. Although some studies reported aspects of salt tolerance mechanisms in sorghum [28,29], to date, the effect of high salt on the germination and the role of exogenous Ca^2+^ to induce increased salt stress tolerance in sorghum have remained elusive.

## 2. Results

### 2.1. Physiological Analysis of the Effect of NaCl and Ca^2+^ on Seed Germination and Growth

In this study we used three parameters including germination assays (germination percentage, mean germination time, germination index, and total germination), measurements of root/shoot length and fresh/dry weights to determine the effects of salt stress and Ca^2+^ on seed germination and growth of *Sorghum bicolor*.

#### 2.1.1. Germination Assays

Germination percentage is described as a qualitative measure that indicates the viability of a population of seeds at a critical stage in the life cycle [30]. *Sorghum bicolor* seeds were germinated in the absence (0 mM) and presence (200 and 300 mM) of NaCl for 7 days (Figure 1). Germination percentage decreased significantly by 30% and 65% for seedlings under 200 mM (*p* ≤ 0.01) and 300 mM (*p* ≤ 0.001) NaCl respectively after day 1. At days 3 and 7, the germination percentage was also significantly (*p* ≤ 0.05) affected by 300 mM NaCl (Figure 1A).

All Ca^2+^ concentrations (5, 15, 35 mM) improved germination of control seedlings (without added NaCl) reaching up to 100% germination percentage (Figure 1B). Only 5 and 15 mM Ca^2+^ increased germination percentage of seeds germinating under 200 mM NaCl on day 1, as compared to the control (200 mM NaCl only), which resulted in a mean germination percentage of ≥70% (Figure 1C). The germination percentage of seedlings germinating under 300 mM NaCl reached up to 56% on day 1 when treated with 5 mM Ca^2+^ (Figure 1D). The highest germination percentage of 100% was observed for control seedlings treated with Ca^2+^, followed by ~98%, which was obtained for seedlings treated with 200 or 300 mM NaCl in the presence of 15 or 5 mM Ca^2+^. Mean germination time, germination index, and total germination have been described in the Appendix A.

#### 2.1.2. Growth Analysis

Root length was measured on day 3 (Appendix A) whereas both roots and shoots were measured on day 7 (Figure 2) and only the results obtained on day 7 will be described in this study. Both the root and shoot lengths significantly (*p* ≤ 0.001) decreased gradually with increasing NaCl concentration by more than 50% relative to the control seedlings (without added NaCl) (Figure 2A). The effect of Ca^2+^ on control seedlings (without added NaCl) was not significant at 5 mM and 15 mM Ca^2+^, whereas high concentrations (35 mM Ca^2+^) significantly (*p* ≤ 0.05) decreased root length (Figure 2B). The effect of low Ca^2+^ (5 and 15 mM) concentrations was not significant on the mean root and shoot lengths of seedlings grown under 200 mM NaCl. But a significant decrease was observed at 35 mM Ca^2+^ for both roots (*p* ≤ 0.001) and shoots (*p* ≤ 0.01).

For seedlings under 300 mM NaCl, the highest root length (15 mm) was observed for the 5 mM Ca^2+^ treatment and the lowest (7.5 mm) was observed for the 35 mM Ca^2+^ treatment as compared to the control (300 mM NaCl only). The same pattern was also observed in the shoots (Figure 2D).

Fresh and dry weights were measured and a reduction was observed under high NaCl (Appendix A). Fresh weights decreased significantly (*p* ≤ 0.001) by 1.7-fold and 1.96-fold for seedlings under 200 and 300 mM NaCl. Control seedlings showed the highest mean fresh weight of 0.57 g and the lowest was 0.29 g for seedlings under 300 mM NaCl. Treatment of control (without added NaCl) and NaCl-stressed seedlings with Ca^2+^ showed no significant effects on the fresh and dry weights.

### 2.2. Biochemical Analysis of the Effect of NaCl and Ca^2+^ on Seed Germination

#### 2.2.1. Proline Accumulation

Proline was measured to determine the capacity of osmotic balance by sorghum in the presence of NaCl and Ca^2+^ (Figure 3). Proline content significantly (*p* ≤ 0.001) increased in the presence of NaCl with no differences observed between 200 and 300 mM NaCl (Figure 3A). Control seedlings (without added NaCl) maintained a low proline content of 106.04 µmol/g^−1^FW as compared to seedlings under high NaCl (300 mM NaCl = 669.25 µmol/g^−1^FW proline). Control seedlings showed a more than 50% increase in proline when supplemented with 15 and 35 mM Ca^2+^ concentrations (Figure 3B). No significant changes in proline content were observed in NaCl-stressed sorghum seedlings treated with different Ca^2+^ concentrations (Figure 3C).

#### 2.2.2. Oxidative Stress

Oxidative damage was determined by assaying the level of hydrogen peroxide (H_2_O_2_) (Figure 4) and membrane structure (Figure 5). H_2_O_2_ content remained at low levels similar to the control for seedlings under 200 mM NaCl, but at 300 mM NaCl, it significantly (*p* ≤ 0.001) increased by approximately 4-fold as compared to the control (Figure 4A). Ca^2+^ had no significant effects on the H_2_O_2_ content of control seedlings (without added NaCl) (Figure 4B). Different Ca^2+^ concentrations significantly (*p* ≤ 0.001) reduced H_2_O_2_ content in seedlings grown under 300 mM NaCl (Figure 4C).

#### 2.2.3. Membrane Structure

The anatomical structure (epidermis and xylem layers) of sorghum seedlings was examined to determine the effect of NaCl and Ca^2+^ on oxidative stress using Scanning Electron Microscopy (SEM). The epidermis of seedlings treated with 5 mM Ca^2+^ revealed a smooth epidermal layer (Figure 5B) as compared to the control (Figure 5A). Under 300 mM NaCl, the epidermis showed changes associated with shrinkage and the formation of several additional features (Figure 5C). The application of 5 mM Ca^2+^ improved the epidermis structure (Figure 5D). Xylem walls of control seedlings in the presence of Ca^2+^ only, showed slight changes (thin layers) (Figure 5F) as compared to the control (Figure 5E). In the presence of NaCl, damage to the xylem is clearly evident as shown by shrinkage and thinning of the walls (Figure 5G). Seedlings under 300 mM NaCl that were treated with 5 mM Ca^2+^ showed improvements e.g., thickened xylem walls (Figure 5H).

#### 2.2.4. Na^+^ and K^+^ Content

Ion content, particularly Na^+^ and K^+^ (Figure 6; Appendix A), was analyzed using Scanning Electron Microscopy-Energy dispersive X-ray spectroscopy *(SEM-EDX)* after treatment with 0 and 300 mM NaCl in the absence and presence of 5 mM Ca^2+^. Na^+^/K^+^ ratio was calculated based on the Weight %, however, the Weight Sigma and Atomic % values are provided in Appendix A. Na^+^ concentration in seedlings treated with 300 mM NaCl increased by 13.5-fold, whereas K^+^ concentration decreased by 1.3-fold resulting in a high Na^+^/K^+^ ratio of 3.19 as compared to the control (Na^+^/K^+^ = 0.17). Treatment with 5 mM Ca^2+^ decreased Na^+^ concentration by 0.8-fold, whereas K^+^ increased by 2.4-fold resulting in a low Na^+^/K^+^ ratio of 1.5. The SEM images revealed significant changes in the morphology of the investigated surface area (Figure 6E–H). Seedlings treated with Ca^2+^ only (Figure 6F) showed a smooth surface area as compared to the control (Figure 6E). In the presence of NaCl, the surface area showed shrinkage (Figure 6G), whereas seedlings treated with both 300 mM NaCl and 5 mM Ca^2+^ showed an improved surface area, with less deformation (Figure 6H).

### 2.3. Transcriptional Analysis of the Effect of NaCl and Ca^2+^ on Germination

The expression of genes encoding the antioxidant, and the sodium/hydrogen exchanger enzymes was analyzed using quantitative real-time PCR (qRT-PCR) in seedlings germinating under 0 mM and 300 mM NaCl with a combination of different Ca^2+^ concentrations (5, 15 and 35 mM) (Figure 7). The antioxidant enzymes included the *Sorghum bicolor* APX2 annotated as the PREDICTED: *Sorghum bicolor* L-ascorbate peroxidase 2, cytosolic (Accession numbers: LOC8077530, mRNA; Sequence ID: XM_002463406.2), and the *Sb*CAT3 annotated as PREDICTED: *Sorghum bicolor* catalase isozyme 3 (Accession numbers: LOC8068221, mRNA Sequence ID, XM 0214460018.1). The Na^+^/H^+^ exchanger antiporter enzymes included the plasma Na^+^/H^+^ exchanger antiporter, also known as the Salt Overly Sensitive (SOS1) annotated as PREDICTED: *Sorghum bicolor* sodium/hydrogen exchanger 8 (Accession numbers: LOC8074408, mRNA Sequence ID: XM 002443629.2), and the vacuolar Na^+^/H^+^ exchanger antiporter annotated as PREDICTED: *Sorghum bicolor* sodium/hydrogen exchanger 2 (Accession numbers: LOC8074408, mRNA Sequence ID: XM 002461123.2). These genes will be referred to as *SbAPX2, SbCAT3, SbSOS1,* and *SbNHX2* in the rest of the document. cDNA amplicons of *ascorbate peroxidase* (*SbAPX2*), *catalase* (*SbCAT3),*
*salt overly sensitive 1* (*SbSOS1*), and *vacuolar Na^+^/H^+^ exchanger antiporter (SbNHX2)* genes, were amplified by qRT-PCR to produce amplicon sizes of 211 bp (*SbAPX2*), 199 bp (*SbCAT3*), 231 bp (*SbSOS1*), and 219 bp (*SbNHX2*) (see Appendix A).

#### 2.3.1. Transcription Analysis of the Antioxidant Genes

Two antioxidant enzymes including APX and CAT were selected based on their ability to detoxify H_2_O_2_ and, therefore, allow the study of effects of NaCl and Ca^2+^ responses by *Sorghum bicolor*. Both genes were constitutively expressed in sorghum seedlings at different levels, with *SbAPX2* representing the highest transcript numbers followed by *SbCAT3* (Figure 7A,B). Their transcript levels increased significantly (*p* ≤ 0.01) in seedlings under 300 mM NaCl (>10-fold relative to their controls) (Figure 7A,B). Treatment of NaCl-stressed seedlings with different Ca^2+^ concentrations reduced their transcripts in a concentration-dependent manner.

#### 2.3.2. Transcription Analysis of the Na^+^/H^+^ (NHX) Antiporter Coding Genes

To understand the effect of NaCl and Ca^2+^ on the ion homeostasis of sorghum, the expression levels of the *plasma membrane Na^+^/H^+^* (*SbSOS1*) and *vacuolar SbNHX2* antiporter genes were determined. Both genes were constitutively expressed, with *SbSOS1* representing the highest transcript level (Figure 7C,D). The *SbSOS1* transcript was significantly (*p* ≤ 0.01) induced in seedlings treated with 300 mM NaCl (~4-fold) relative to the control. The presence of different Ca^2+^ concentrations decreased the expression of *SbSOS1* transcript in a concentration-dependent manner (Figure 7C). The expression of the *vacuolar SbNHX2* was not significantly affected by 300 mM NaCl, whereas treatment with different Ca^2+^ concentrations increased its expression significantly (*p* ≤ 0.01), with a 65-fold increase under 35 mM Ca^2+^ relative to the control (Figure 7D).

## 3. Discussion

Salt stress greatly affects many biological processes including seed germination by inducing a reduction in the germination rate and delayed initiation of germination and seedling establishment [31,32]. In this study, the effects of NaCl stress and Ca^2+^ on the germination of *Sorghum bicolor* were investigated. Salt stress reduced germination percentage and delayed germination of sorghum seeds with significant effects observed at 300 mM NaCl within the first 3 days (Figure 1A). Similar effects were reported in *Festuca ovina* L. [24], *Cucumis sativus* L. cv. [25], *Brassica napus* L. [33], *Suaeda salsa* [34], and even a halophyte, *Halocnemum strobilaceum* [35]. The germination percentage assays are supported by the germination index, which decreased from 136 to 95.66 under 300 mM NaCl, indicating a ~30% reduction (Appendix A). Previous reports have also demonstrated that increasing NaCl concentrations can affect the germination index of crops as observed in *Zea mays* [36] and *Capsicum annuum* L. [37]. The decreased germination index might be due to osmotic stress, which causes impairment in nutrient uptake and ionic stress due to the accumulation of ions causing ion toxicity [20,38]. Application of Ca^2+^ was able to reverse the effects of salt stress on sorghum seed germination especially at low concentrations (5 mM Ca^2+^). Similarly, Ca^2+^ also improved seed germination of *Festuca ovina* L. [24], *Phragmites karka* [39], *Triticum aestivum* [40], *Urochondra setulosa* [41], *Pisum sativum* [42], *Solanum lycopersicon* [43,44], and *Vigna radiata* [45].

Sodium chloride also reduced root and shoot length of sorghum seedlings in a concentration-dependent manner and shoots showed more sensitivity to NaCl than roots (Figure 2). This is possibly due to excess salts in the roots causing ionic stress and a decrease in root osmotic potential that prevents the roots from absorbing water and water transport to the shoots. This would in turn affect embryo expansion and seedling emergence [46]. Similar effects were observed in *Brassica juncea*, which showed decreased growth and seedling emergence due to salt stress [38]. Ca^2+^ on its own decreased root length of seedlings grown in the absence (0 mM NaCl) and presence (200 mM NaCl) of salt with significant effects observed for higher Ca^2+^ (35 mM) concentrations. However, Ca^2+^ improved both root and shoot length of seedlings grown in the presence of 300 mM NaCl, and these responses are similar to those seen in *Festuca ovina* L. [24]. The highest root and shoot lengths were observed in seedlings that were treated with 5 or 15 mM Ca^2+^, followed by a decrease at high Ca^2+^ concentration (35 mM). Given these observations, it was important to determine the Ca^2+^ concentrations best suitable to improve growth without causing toxicity and inhibitory growth effects [47], thus 5 and 15 mM Ca^2+^ represent the best analyzed in this study. Increasing salt concentration decreased fresh and dry weights of sorghum seedlings indicating that cell division and elongation were inhibited [48,49]. Treatment with Ca^2+^ showed no significant improvement in the fresh and dry weight of seedlings germinated in the presence of NaCl (Appendix A). Although the effect of Ca^2+^ on fresh and dry weights of NaCl-stressed sorghum seedlings was not significant, other studies reported positive effects [24,50].

Proline is an important osmolyte that helps to maintain the osmotic balance by increasing the osmotic potential during osmotic stress [13]. Sorghum seedlings accumulated high proline content under NaCl stress (Figure 3A). Although at different stages, these results are supported by other studies, which reported similar effects in sorghum [29,51,52] and other crops [53]. A high concentration of Ca^2+^ (15 and 35 mM) significantly increased proline content in control seedlings (Figure 3B), whereas in NaCl treated seedlings, there were no significant changes (Figure 3C). This suggests that the protective effect of Ca^2+^ on sorghum might be independent of proline regulation. Since proline also acts as a molecular chaperone, protects photosynthesis, antioxidant enzymes, and prevent membrane damage [13,54]. This might be the reason for its stable accumulation under NaCl stress and in combination with Ca^2+^ (Figure 3C), which can be linked to the induction of *pyrroline-5-carboxlyate synthetase* 1 (*P5CS1*), a key enzyme for proline biosynthesis that is dependent on Ca^2+^ signaling [55,56]. The results further suggest that sorghum seedlings adapt their osmotic potential under NaCl stress, consistent with the high accumulation of osmolytes.

Sodium chloride stress causes oxidative stress through the accumulation of ROS, which can affect membrane structure and function as well as enzyme activities and with it, metabolic processes [57]. A significant accumulation of H_2_O_2_ in seedlings treated with 300 mM NaCl was observed and these levels were reduced by supplementation with Ca^2+^ (Figure 4A,C). Salt causes membrane damage due to oxidative stress and does so by displacing Ca^2+^ from the phospholipid membrane binding sites [7]. However, this effect can be reversed, at least in parts, by the addition of Ca^2+^, which affects the uptake and transport of ions and helps to maintain membrane integrity [19,24,57]. In this study, NaCl triggered damage to the epidermis and xylem layers of sorghum seedlings as shown by the formation of structures and shrinkage on these layers (Figure 5C,G). Ca^2+^ partly stabilized the lipid bilayer of cellular membranes and provided structural integrity under these conditions (Figure 5H). This is supported by the smoothened epidermis and the thickened xylem layers of seedlings treated with both NaCl and Ca^2+^ (Figure 5D,H). These results indicate that Ca^2+^ can effectively alleviate NaCl-induced oxidative stress as previously reported [58,59].

Salt stress is also responsible for the ionic imbalance due to the accumulation of excess Na^+^ and a decrease of K^+^ and Ca^2+^ [19,60]. In this study, this was evident by the high Na^+^/K^+^ ratio of 3.19 under 300 mM NaCl, and this effect was reversed by the exogenous application of 5 mM Ca^2+^ resulting in a low Na^+^/K^+^ ratio of 1.5 (Figure 6C,D). This is consistent with the effective maintenance of ion homeostasis by Ca^2+^. Salt stress causes membrane damage due to oxidative stress by displacing Ca^2+^ from the phospholipid membrane binding sites [7]. This is true since seedlings treated with NaCl revealed shrinkage and some deformation as compared to the control (Figure 6G). However, this effect was reversed, at least in parts, by the addition of Ca^2+^ (Figure 6H), which affects the uptake and transport of ions and helps to maintain membrane integrity [19,24,57].

The antioxidant system responsible for scavenging ROS was activated (Figure 4A–C) as seen by the induced expression of antioxidant genes, *ascorbate peroxidase (SbAPX2) and catalase (SbCAT3)* under NaCl stress (Figure 7A,B). H_2_O_2_ is regulated by various antioxidant enzymes, which reduce it to water. APX and CAT are considered the most important enzymes in the detoxification of H_2_O_2_ [61,62]. *Sb**APX2* and *SbCAT3* genes were constitutively expressed but their expression was highly induced under 300 mM NaCl (Figure 7A,B). Thus, the increased expression of *APX* and *CAT* indicates a protective mechanism since these genes are transcribed and translated into enzymes and hence, enable detoxification of excess H_2_O_2_. Additionally overexpression of the *APX* gene in plants is associated with improved protection against oxidative stress [63]. Similarly,, microarray data indicated an increase in the expression of *CAT* and *APX* in *Arabidopsis thaliana* under abiotic (salinity: 100 mmol/L NaCl, cold: 10 °C, heat: 38 °C and light: 800 μmol photons m^−2^s^−1^) stresses [64]. The expression of *APX* was also increased in *Oryza sativa* [65], whereas that of *CAT* was also increased in *Lotus japonicus* [66] and *Cuminum cyminum* L. [67] in response to 50, 100, and 150 mM NaCl. Although transcriptional levels do not necessarily always correlate with protein level or enzyme activity, other studies have reported on the increased activity of antioxidant enzymes under NaCl stress in germinating seeds [68,69]. This expression occurred concomitantly with effective ROS scavenging and reduction of oxidative stress in seedlings under stress. Ca^2+^ significantly decreased the transcript levels of *SbAPX2* and *SbCAT3* under salt conditions in a concentration-dependent manner (Figure 7A,B). The expression of *SbAPX2* was decreased to the same magnitude as that of the control, indicative of significant stimulation of detoxification by Ca^2+^. APX has a higher affinity (µM range) for H_2_O_2_ than CAT (mM range) [70], which may explain why *SbAPX2* transcript levels were about ~7-fold higher than those of *SbCAT3* in seedlings treated with 300 mM NaCl, despite the fact that both enzymes are crucial for H_2_O_2_ detoxification.

The expression of the antiporter genes, *SOS1,* and *vacuolar NHX2* that are responsible for maintaining ion homeostasis [71,72] were measured. *SbSOS1* was upregulated under 300 mM NaCl stress in sorghum seedlings (Figure 7C). This may be explained by its role in the exclusion of toxic Na^+^ into the root apoplast region away from the cells delaying uptake into shoots and leaves [14,16]. Overexpression of *SOS1* was associated with increased salt tolerance in different species [17], suggesting that expression of *SbSOS1* in this study might have been sufficient to prevent ion toxicity and confer increased NaCl tolerance. Both, the constitutive and stress-inducible expression of the sorghum *SOS1* in this study show a similar pattern to the expression of *SOS1* in Brassica and wheat genotypes [73,74,75]. In the presence of Ca^2+^, the expression of *SbSOS1* decreased in a concentration-dependent way. Thus, we propose that the application of exogenous Ca^2+^ induce tolerance maybe through binding of Ca^2+^ to the phospholipid bilayer of membranes to prevent the uptake and transport of Na^+^ into cells [76,77]. In turn, this may partly eliminate the need for strong *SOS1* expression. The *vacuolar SbNHX2* was constitutively expressed in germinating seedlings, but the transcript level was reduced by salt stress (Figure 7D). Incidentally, similar results were observed previously [78,79,80]. However, most studies reported an upregulation of other *NHX* genes under salt stress [72,81,82,83,84,85,86]. To our surprise, *vacuolar SbNHX2* was significantly induced by a combination of 300 mM NaCl and 35 mM Ca^2+^. These results suggest that the sorghum vacuolar NHX2 has a role in stress responses that might be mediated by Ca^2+^. Overall gene expression of both the antioxidant and the *SOS1* genes in sorghum showed a strong correlation with the alleviation of oxidative stress caused by ROS accumulation and this may be linked to Ca^2+^ signaling. The structural, physiological, and biochemical role of Ca^2+^ in *Sorghum bicolor* await further elucidation.

## 4. Conclusions

This study illustrated the detrimental effects of salt stress on *Sorghum bicolor* and the ability of Ca^2+^ to ameliorate these effects. Sorghum is known to be moderately tolerant to stress and this was further confirmed in this study since seeds reached up to 98% germination percentage under high salt (300 mM NaCl). This indicated that the crop has some level of tolerance, which is also due to osmolyte accumulation even in the presence of Ca^2+^. Sorghum has a strong antioxidant and ion homeostasis system, as suggested by the high H_2_O_2_ content, Na^+^/K^+^ ratio, and transcript levels (*SbAPX2, SbCAT,3* and *SbSOS1* genes), all of which were reduced by supplementation with Ca^2+^. This resulted in improved seedlings growth and protection of membrane structures. Therefore, the study proposes that 5 mM Ca^2+^ effectively alleviated the effects of salt stress without causing any severe negative effects. Finally, it will be necessary to conduct further analysis in the field in order to broaden our understanding of how to improve the growth of this important crop under salt stress with supplementation of Ca^2+^.

## 5. Materials and Methods

### 5.1. Seed Preparation and Growth Conditions

Red Sorghum (*Sorghum bicolor*) seeds were purchased from Agricol, Brackenfell Cape Town, South Africa. In order to maintain uniformity in the study, seeds were chosen based on possessing an identical size and color. After surface decontamination [87], seeds were imbibed overnight in distilled H_2_O at 25 °C with shaking at 600 rpm. Seeds were dried under the laminar flow and five seeds were sown on filter paper placed on sterilized Petri dishes containing 4 mL of solutions. The solutions included distilled H_2_O only (0 mM), 200 mM and 300 mM sodium chloride and different CaCl_2_ (Ca^2+^) concentrations (5 mM, 15 mM, and 35 mM). Petri dishes were placed at 25 °C and seeds germinated for 7 days in the dark. Seeds were inspected daily and the number of germinated seed (> 2 mm radicle) weascounted. On day 3 only, the roots were measured, whereas both roots and shoots were measured on day 7. Seedlings were harvested on day 7, rinsed thoroughly and used immediately or stored at −80 °C for future use.

### 5.2. Physiological analysis

Germination assays including germination percentage, mean germination time, germination index, and total germination were calculated according to the equations below:

#### 5.2.1. Germination Assays

Four germination parameters including germination percentage, mean germination time, germination index and total germination were measured using formulas shown below:

Germination percentage (GP): n/N × 100, where n: total number of seeds germinated and N: the total number of seeds sown [38].

Mean Germination Time (MGT): ∑ f * x/ ∑ f, where f: number of seeds germinated on day x. [30].

Germination index (GI): ∑ (n1 × d7) + (n2 × d6) + (n3 × d5)……(n7 × d7) where n1: number of seeds germinated on day 1 and d7: number of seeds germinated on day 7. According to Kader 2005 where a heavier weight is given to seeds germinated on the first day and less weight given to those germinated on the last day.

Total germination (TG): d7/N × 100, where d7 total number of seeds germinated on the final day (day 7) and N: the number of seeds germinated [30].

#### 5.2.2. Growth Attributes

Seedling lengths were measured with a ruler (mm scale), roots were measured at day 3, whereas both root and shoot length were measured at day 7. Dry weights were obtained after oven-drying seedlings at 80 °C overnight or until constant weight.

### 5.3. Biochemical Analysis

#### 5.3.1. Proline Content

Proline content was determined as previously described [38], with slight modifications. About 0.5 g of grounded plant material was homogenized with 2 mL of sulfosalicylic acid. The homogenized samples were then centrifuged at 5000 rpm for 5 min and 0.5 mL of the supernatant was transferred to a clean tube containing 0.5 mL acetic acid and 0.5 mL ninhydrin acid reagent (1.25 g of ninhydrin in 30 mL of 99% acetic acid and 20 mL of 6 M H_3_PO_4_) and boiled for 45 min in a water bath. Samples were then allowed to cool on ice completely before adding equal volumes of toluene and reading the optical density of the samples at 520 nm using a spectrotometer Helios^®^ Epsilon visible 8nm bandwidth (Thermo Fisher Scientific, USA).

#### 5.3.2. Hydrogen Peroxide Content

Hydrogen peroxide (H_2_O_2_) was assayed following the optimized method by Junglee et al. [88]. About 0.15 g ground plant material was homogenized with 0.25 mL TCA (0.1% *w/v*), 0.5 mL potassium iodide (1 M), and 0.25 mL potassium phosphate buffer (10 mM, pH 6). Tubes were vortexed and centrifuged for 15 min at 10,000 rpm (4 °C). Samples were transferred to a 96 well microtiter plate and were allowed to incubate at room temperature for 20 min. Absorbance was read at 390 nm using the FLUOstar^®^ Omega (BMG LABTECH, Ortenberg, Germany) microtiter plate reader. H_2_O_2_ was quantified by generating a standard curve with an H_2_O_2_ solution.

#### 5.3.3. Anatomical and Element Analysis

Element and anatomical analysis were done using the Scanning Electron Microscopy-Energy dispersive X-ray spectroscopy *(SEM-EDX)* system that is located at the electron microscope unit, Physics Department, University of the Western Cape. Analysis was undertaken of *Sorghum bicolor* root seedlings that were grown in the absence (0 mM) and presence (300 mM) of NaCl and treated with 5 mM CaCl_2_. Samples were placed on aluminum stubs coated with conductive carbon tape. The samples were then coated with a thin layer of carbon using an EMITECH-K950x carbon coater. All EDX spectra were collected with an Oxford X-Max silicon solid-state drift detector at an accelerating voltage of 20 kV for 60 s to ensure proper x-ray detection. All spectra were analyzed using the build-in Oxford Aztec software suite. Samples were then imaged and images collected using a Zeiss Auriga field emission gun scanning electron microscope, operated at an accelerating voltage of 5 kV using an in-lens secondary electron detector.

### 5.4. Quantitative Real-Time PCR

The cDNA that was used as the template for the quantitative real-time PCR (qRT-PCR) experiment was synthesized as previously described [87]. Quantitative RT-PCR was used to analyze the expression profiles of sorghum antioxidants (*SbAPX2* and *SbCAT3*) and the Na^+^/H^+^ exchanger antiporter *(SbSOS1* and *vacuolar SbNHX2)* genes. Reference genes included Beta-actin and Ubiquitin as previously described [24], the sequences and accession numbers of the target genes can be found in Appendix A. The reaction mixture contained 1 µL template cDNA, 5 µL 2x SYBR Green I Master Mix (Roche Applied Science, Upper Bavaria, Germany), varying concentrations of each primer, and ddH_2_O added to a final volume of 10 µL. The reactions were subjected to 95 °C for 10 min, 45 cycles at 95 °C for 10 s, 55 °C (*SbAPX2*; *SbCAT3*); 58 °C (*SbSOS1*) and 60 °C (*SbNHX2*); for 10 s, and 72 °C for 20 s. A melting curve analysis was also performed using default parameters on the LightCycler^®^ 480 instrument (Roche Applied Science, Upper Bavaria, Germany). The expression levels of the target genes were normalized to the reference genes and analyzed using the LightCycler^®^ 480 SW (version 1.5) data analysis software. The expression was quantified by the relative quantification method using a standard curve of serially diluted cDNA templates. Each qRT-PCR reaction was done in triplicate and three non-template controls were included. Data represent the average of three independent experiments.

### 5.5. Statistical Analysis

All experiments were repeated at least four times and data were statistically analyzed by the two-way ANOVA using GraphPad Prism 8.4.2 (https://www.graphpad.com). Data in Figures and Tables represent the mean ± standard deviation. Statistical significance between control and treated plants were determined by the Bonferroni’s multiple comparison test and represented as *** = *p* ≤ 0.001, ** = *p* ≤ 0.01, and * = *p* ≤ 0.05. Appendix A for all analyses have been supplied including physiological and biochemical analysis raw data (Appendix A), Sequencing data (Appendix A) and gene expression data (Appendix A).

## Figures and Tables

**Figure 1 plants-09-00730-f001:**
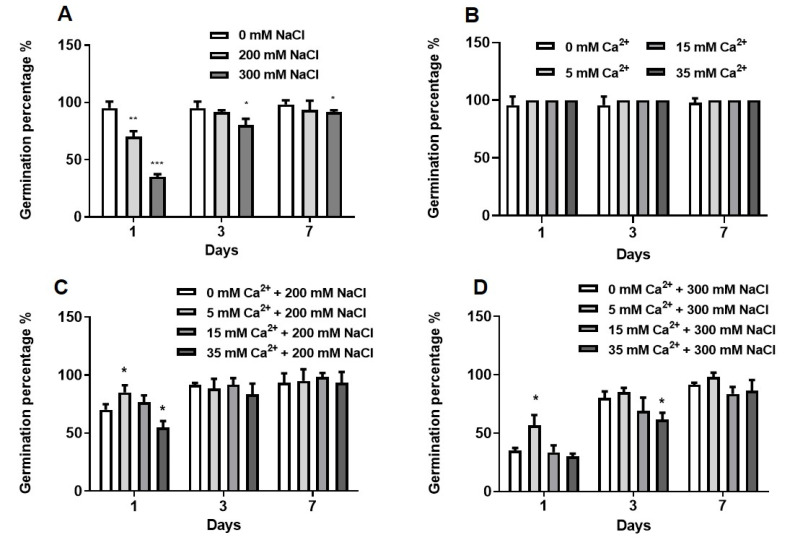
Effect of NaCl stress and Ca^2+^ on the germination percentage of sorghum seedlings. (**A**) Seedlings germinated under different NaCl concentrations only. (**B**–**D**) Seedlings germinated under different NaCl and Ca^2+^ (5, 15 and 35 mM) concentrations, (**B**) 0 mM, (**C**) 200 mM NaCl and (**D**) 300 mM NaCl. Error bars represent the SD calculated from three biological replicates. Statistical significance between control and treated plants was determined using two-way ANOVA conducted on GraphPad Prism 8.4.2, shown as *** = *p* ≤ 0.01, ** = *p* ≤ 0.01, and * = *p* ≤ 0.05 according to the Bonferroni’s multiple comparison test.

**Figure 2 plants-09-00730-f002:**
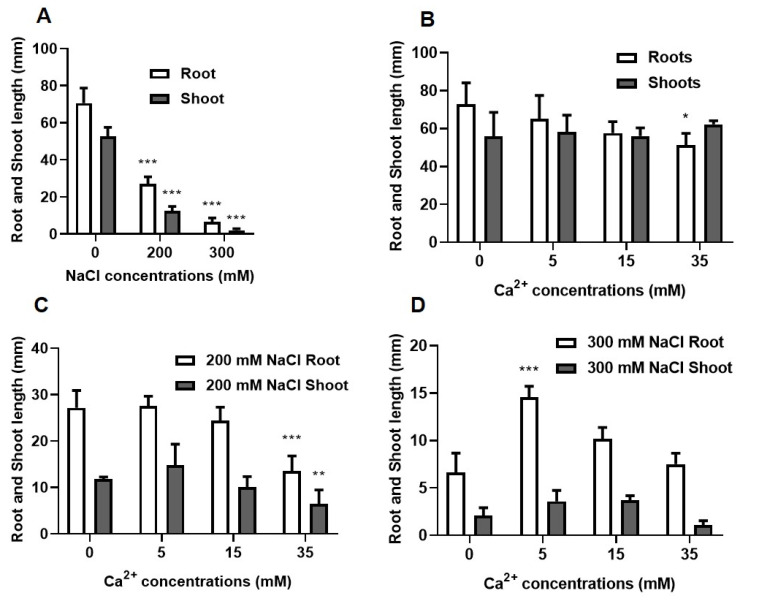
Effect of NaCl and Ca^2+^ on root and shoot length of sorghum seedlings measured on day 7. (**A**) Seedlings germinated in the presence of different NaCl concentrations only. (**B**–**D**) Seedlings germinated under different NaCl and Ca^2+^ (5, 15 and 35 mM) concentrations; (**B**) 0 mM, (**C**) 200 mM and (**D**) 300 mM NaCl. Error bars represent the SD calculated from three biological replicates. Statistical significance between control and treated plants was determined using two-way ANOVA conducted on GraphPad Prism 8.4.2, shown as ** = *p* ≤ 0.01, and * = *p* ≤ 0.05 according to the Bonferroni’s multiple comparison test.

**Figure 3 plants-09-00730-f003:**
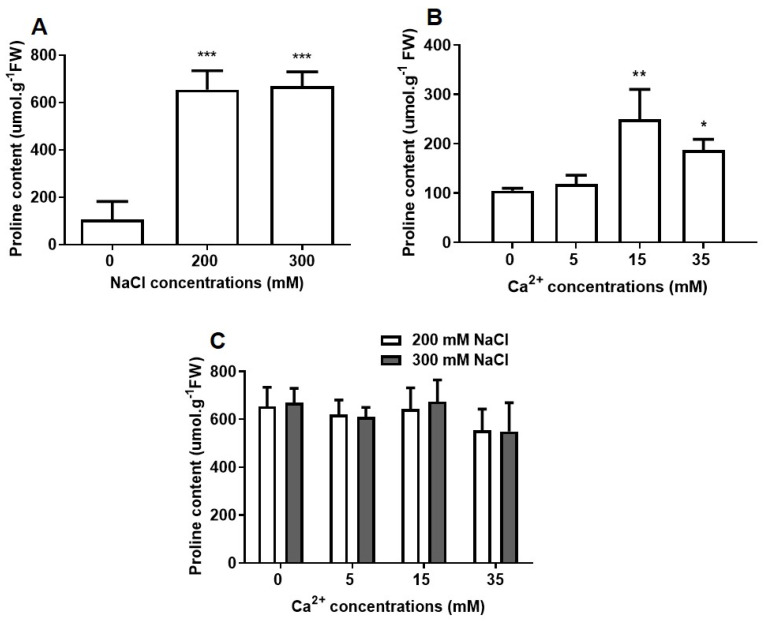
Effect of NaCl and Ca^2+^ on proline accumulation in sorghum seedlings. Proline content measured on (**A**) seedlings germinated in the presence of different NaCl concentrations only. (**B**,**C**) Seedlings germinated under different NaCl and Ca^2+^ (5, 15 and 35 mM) concentrations; (**B**) 0 mM, (**C**) 200 mM and 300 mM NaCl. Error bars represent the SD calculated from three biological replicates. Statistical significance between control and treated plants was determined using two-way ANOVA conducted on GraphPad Prism 8.4.2, shown as *** = *p* ≤ 0.001, ** = *p* ≤ 0.01, and * = *p* ≤ 0.05 according to the Bonferroni’s multiple comparison test.

**Figure 4 plants-09-00730-f004:**
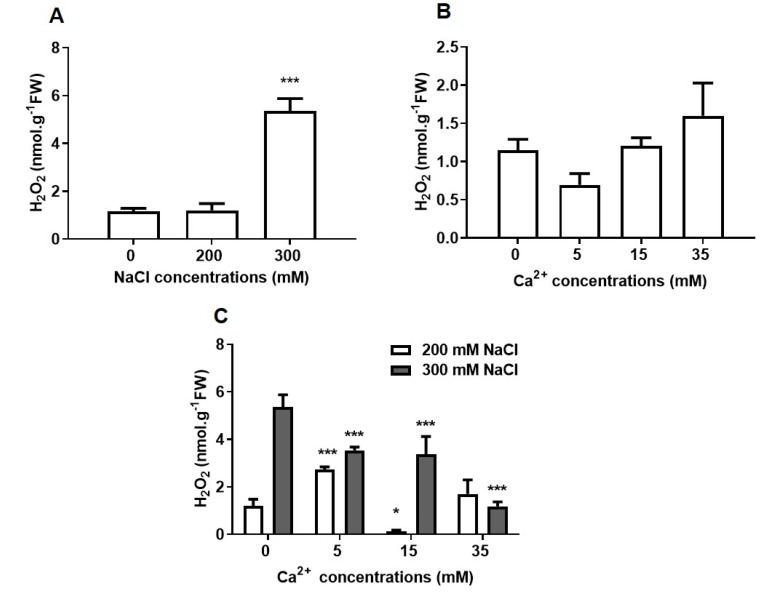
Effect of NaCl and Ca^2+^ on H_2_O_2_ content of sorghum seedlings. (**A**) H_2_O_2_ content measured from seedlings germinated in the presence of different NaCl concentrations only. (**B**,**C**), Seedlings under different NaCl and Ca^2+^ (5, 15 and 35mM) concentrations; (**B**) 0 mM, (**C**) 200 and 300 mM NaCl. Error bars represent the SD calculated from three biological replicates. Statistical significance between control and treated plants was determined using two-way ANOVA conducted on GraphPad Prism 8.4.2, shown as *** = *p* ≤ 0.001 and * = *p* ≤ 0.05 according to the Bonferroni’s multiple comparison test.

**Figure 5 plants-09-00730-f005:**
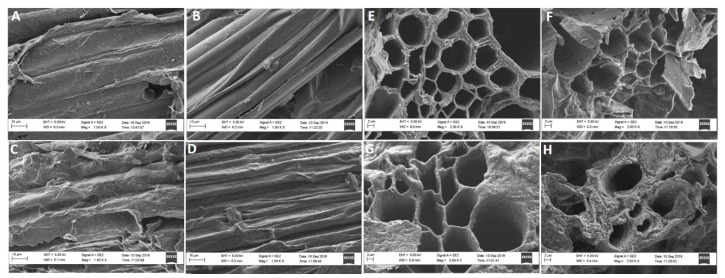
Anatomical image showing the epidermis and xylem layers of *Sorghum bicolor* analyzed using Scanning Electron Microscopy. Cross section of epidermis layer of seedlings under (**A**) 0 mM NaCl, (**B**) 0 mM NaCl + 5 mM Ca^2+^, (**C**) 300 mM NaCl, (**D**) 300 mM + 5 mM Ca^2+^. Cross section of the xylem layers of seedlings under (**E**) 0 mM NaCl, (**F**) 0 mM NaCl + 5 mM Ca^2+^, (**G**) 300 mM NaCl, (**H**) 300 mM + 5 mM Ca^2+^.

**Figure 6 plants-09-00730-f006:**
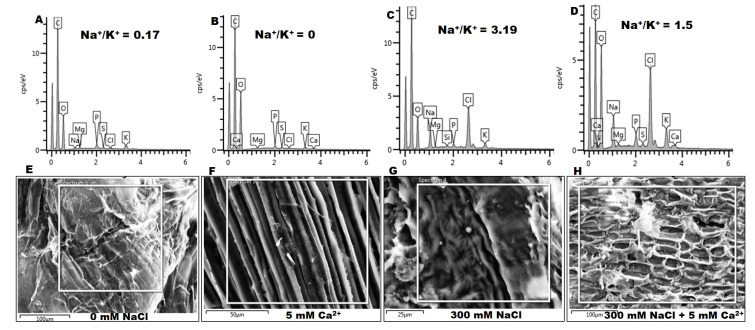
Energy dispersive X-ray (EDX) spectroscopy and SEM images of the effect of NaCl and Ca^2+^ on the concentration of Na^+^ and K^+^ of sorghum seedlings. (**A**–**D**) Spectra showing concentration of different elements, (**E**–**H**) Sorghum surface for mapped elements. (**A**,**E**) Na^+^ and K^+^ content in the absence of NaCl (0 mM), (**B**,**F**) presence of 5 mM Ca^2+^ only, (**C**,**G**) 300 mM NaCl only, (**D**,**H**) and 300 mM + 5 mM Ca^2+^.

**Figure 7 plants-09-00730-f007:**
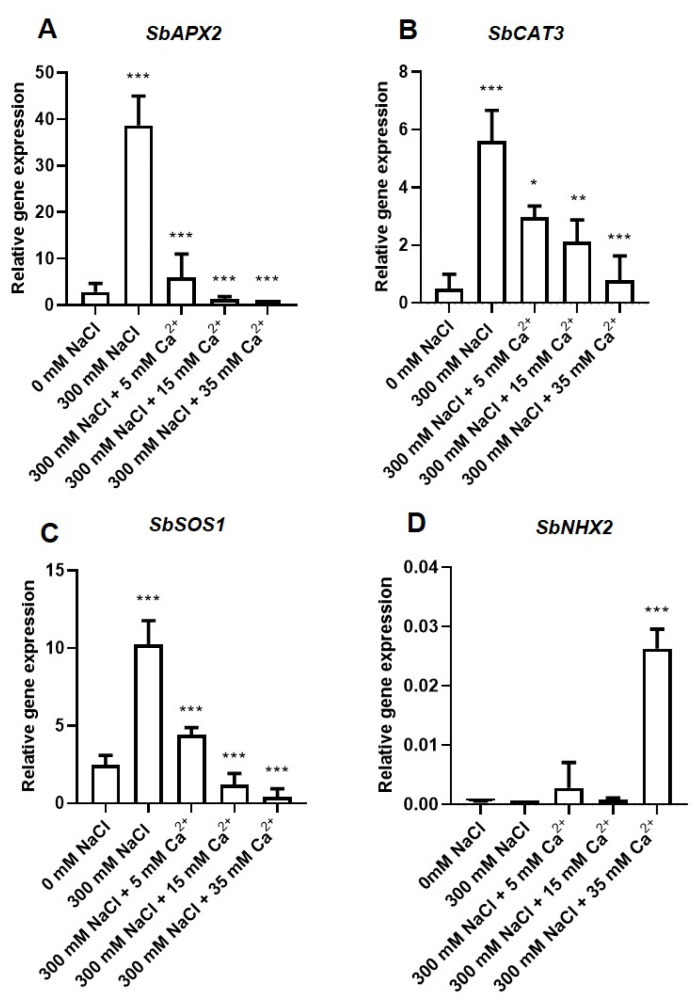
Transcript analysis of *Sorghum bicolor* antioxidant and Na^+^/H^+^ exchanger antiporter genes in the absence (0 mM) and presence (300 mM) of NaCl treated with different Ca^2+^ (5, 15, and 35 mM) concentrations. Relative gene expression of (**A**) *SbAPX2*, (**B**) *SbCAT3*, (**C**) *SbSOS1* and (**D**) *vacuolar SbNHX2*. Error bars represent the SD calculated from three biological replicates. Statistical significance between control and treated plants were determined by two-way ANOVA conducted on GraphPad Prism 8.4.2, shown as ** = *p* ≤ 0.01, and * = *p* ≤ 0.05 according to the Bonferroni’s multiple comparison test.

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
