# Peer review of "Calcium Improves Germination and Growth of *Sorghum bicolor* Seedlings under Salt Stress"

_plants, 2020, doi:10.3390/plants9060730_

Round 1
Reviewer 1 Report
The manuscript entitled “Calcium improves germination and growth of Sorghum bicolor seedlings under salt stress” by Mulaudzi-Masuku et al. tried to investigate how moderate Ca2+ mitigated salinity stress in sorghum. Below are some comments to improve the quality of the manuscript.
Some comments:
- Low concentration of Ca2+ mitigates salinity stress in plants is well known. Showing the routine work/results may not excite the readers. Authors may consider to reschedule the manuscript to focus on the most impressive results.
- Besides the phenotypic results, no further experimental evidences were provided for the Ca2+ improved seed germination of sorghum under salt stress. I suggest to move the germination results to the supplementary file.
- Why measuring bulk Na+ K+ with the SEM-EDS imaging technique? How the background line was set? How was the noise level? For example, the Na+ peak can be barely seen in the Figure 6A. With the possible noise, it is hard to calculate the Na+ level. Also, the SEM-EDX images for each looked ions need to be provided.
- Figure 7. APX and CAT are isozymes. More details need to be added regarding the accurate identity of the investigated APX and CAT genes. Also, NHX is a family name of Na+/H+ exchanger. So, be more specific on the investigated genes.
- Figure 5 is more impressive for the readers. More efforts could be focused on this topic. This will make the story of this manuscript more different with the previous papers.
- There are several citations from 1980s. Authors may consider to update it with more recent one.
Author Response
Reviewer 1: Comments and Suggestions for Authors
The manuscript entitled “Calcium improves germination and growth of Sorghum bicolor seedlings under salt stress” by Mulaudzi-Masuku et al. tried to investigate how moderate Ca2+ mitigated salinity stress in sorghum. Below are some comments to improve the quality of the manuscript.
Some comments:
- Low concentration of Ca2+ mitigates salinity stress in plants is well known. Showing the routine work/results may not excite the readers. Authors may consider to reschedule the manuscript to focus on the most impressive results.
Response: We are aware of the effect of low Ca2+ concentration and have therefore moved most of the statistically insignificant results to the supplementary file as indicated below in 2.2.
2.1 Besides the phenotypic results, no further experimental evidences were provided for the Ca2+ improved seed germination of sorghum under salt stress.
Response: The biochemical, including measurement of H2O2 content, membrane structure and the ion content (Section 2.2), and transcriptional (Section 2.3) analysis are evidences that support the role of Ca2+ to improve germination under salt stress in sorghum.
2.2 I suggest to move the germination results to the supplementary file.
Response:
The entire work is based on the germination of sorghum, thus we believe that it is important to leave at least one germination assay (i.e germination percentage) in the main document while the rest of the germination assays (mean germination time, germination index and total germination) including Table 1, have been moved to supplementary file. In addition supplementary figures, Figure S1, S2 and S3 have been deleted since this information can also be obtained from Table S1.
3. Why measuring bulk Na+ K+ with the SEM-EDS imaging technique?
Response: The analysis was conducted for a larger number of different elements as described previously (Ragavendran et al., Element analysis of Aerva lanata (L.) by EDX method. International Research Journal of Pharmacy 2012, 3 (7) 218-220).
However, for the purpose of this study only Na+ and K+ are shown.
3.1 How the background line was set? How was the noise level? For example, the Na+ peak can be barely seen in the Figure 6A. With the possible noise, it is hard to calculate the Na+
Response: Detailed explanation of how the measurements were conducted is included under methods section 5.3.3.
In addition it is possible that the Na+ peak is not very distinct in Figure 6A, due to the fact that the medium used was dH2O only.
The raw data for the specific values of each measured ion is also provided in Table S3.
Furthermore, the built-in software does the analysis and the background models are complicated and one has to study them in detail to get a proper model to fit the background. The signal-to-noise levels are important only if one does not get proper x-ray counts and the spectra are not smooth (which was not the case in our samples).
3.1 Also, the SEM-EDX images for each looked ions need to be provided.
Response: The SEM-EDX images for each treatment rather than individual ions have been provided in Figure 6E-H.
4. Figure 7. APX and CAT are isozymes. More details need to be added regarding the accurate identity of the investigated APX and CAT genes. Also, NHX is a family name of Na+/H+ exchanger. So, be more specific on the investigated genes.
Response: An extended description of the genes analysed has been added in page 12, Section 2.4, Lines 259-266 and page 13, Lines 267-275.
Page 13, Lines 277-278, 279-282, the figure legends and the figure have been edited.
5. Figure 5 is more impressive for the readers. More efforts could be focused on this topic. This will make the story of this manuscript more different with the previous papers.
Response: Authors have provided adequate information in the results and discussion sections.
6. There are several citations from 1980s. Authors may consider to update it with more recent one.
Response:
The following changes were implemented in the reference section:
Page 23, line 593-594 [7] was replaced with “Yermiyahu, U.; Nir, S.; Ben-Hayyim, G.; Kafkafi, U.; Kinraide, T.B. Root elongation in saline solution related to calcium binding to root cell plasma membranes. Plant and Soil. 1997, 67-76.”
Page 23, line 612-613 deleted reference [14] Clarkson, D.T.; Hanson, J.B. The mineral nutrition of higher plants. Annu. Rev. Plant Physiol. 1980, 31, 239–298.
Page 26, lines 718-721 deleted references [59] and [60], [59] Leopold, A.; Willing, R. Evidence for toxicity effects of salt on membranes, ‘In Salinity Tolerance in Plants’ Wiley & Sons: New York, NY, USA, 1984; 67-76.
[60] Zhao, K.; Mingliang, L. Alleviating NaCI induced injurious effects by calcium. Proc. Internat. Congr. Plant Physiol. 1988, 15–20.
References [50] and [60] were deleted and replaced with the new reference [7]
Page 27, line 774-775, deleted reference [80] Kurth, E.; Cramer, G.R.; Läuchli, A.; Epstein, E. Effects of NaCl and CaCl(2) on cell enlargement and cell production in cotton roots. Plant Physiol. 1986, 82, 1102–1106.

Reviewer 2 Report
The manuscript were well revised so I believe this manuscript is acceptable for publication.
Author Response
Reviewer 2: Comments and Suggestions for Authors
The manuscript were well revised so I believe this manuscript is acceptable for publication.
Response: We would like to thank the reviewer for the above comment.

Reviewer 3 Report
The manuscript entitled "Calcium improves germination and growth of Sorghum bicolor seedlings under salt stress", which is the resubmitted version of plants-579924 is much better written now. I see a significant improvement in the methodological approach, presentation of results, discussion and the references. The text currently submitted is concise and much more easier to read. However, in such a refined correction some bad impression leaves the use of term "sterilization" regarding surface decontamination of Sorghum bicolor seeds (line 425). Congratulations, on the results of the author's work.
Author Response
Reviewer 3: Comments and Suggestions for Authors
The manuscript entitled "Calcium improves germination and growth of Sorghum bicolor seedlings under salt stress", which is the resubmitted version of plants-579924 is much better written now. I see a significant improvement in the methodological approach, presentation of results, discussion and the references. The text currently submitted is concise and much easier to read. However, in such a refined correction some bad impression leaves the use of term "sterilization" regarding surface decontamination of Sorghum bicolor seeds (line 425). Congratulations, on the results of the author's work.
Response: We implemented the suggestion and changed the term from “sterilization” to “surface decontamination” in Page 17, Line 430.

Round 2
Reviewer 1 Report
Authors largely failed to properly respond to the main points of the last round review report. The current manuscript does not add much scientific information to the research community of plant salinity stress tolerance. Also, the SEM-EDX data lack the credibility. The explanation authors given in the rebuttal letter are not convincing. Overall, this manuscript may be more suitable for other journals.
Author Response
Reviewer 1: Comments and Suggestions for Authors: Authors largely failed to properly respond to the main points of the last round review report. The current manuscript does not add much scientific information to the research community of plant salinity stress tolerance. Also, the SEM-EDX data lack the credibility. The explanation authors given in the rebuttal letter are not convincing. Overall, this manuscript may be more suitable for other journals.
Response:
- We do not agree with the assessment of the Reviewer on the novelty of the MS and we think that doubting the credibility of the data in the absence of technical criticism is inappropriate.
Finally we thank all the reviewers and the editor for the effort that they have put in improving our manuscript.
This manuscript is a resubmission of an earlier submission. The following is a list of the peer review reports and author responses from that submission.
Round 1
Reviewer 1 Report
The manuscript entitled “Calcium improves germination and growth of Sorghum bicolor seedlings under salt stress” investigated the role of Ca2+ in alleviating salt stress in sorghum germination and growth and tried to link it with antioxidant system. However, Ca2+ mitigated salinity stress in plants is well known. Also, the role of antioxidant system in Ca2+ mitigated salinity stress in plants is also known. Authors only used a different plant species. Lacking the novel mechanistic insight into Ca2+ improved seed germination and plant growth is a drawback of the current manuscript.
Comments:
Why only looked at the genes of SOS pathway? SOS pathway is important for Na+ extrusion. However, besides Na+ extrusion, vacuolar Na+ sequestration and cell’s ability to maintain K+ are also important mechanisms for salt stress tolerance. Sorghum bicolor seedlings start showing a significant decline of FW after 100 mM NaCl stress. However, H2O2 content only increased in the seedlings under 300 mM NaCl. 4B. Compared with no CaCl2 treatment, adding 5 mM CaCl2 increased H2O2 content of Sorghum bicolor seedlings under 200 mM NaCl. However, the 15 mM CaCl2 addition significantly reduced H2O2 in Sorghum bicolor seedlings under 200 mM NaCl compared with no CaCl2 treatment. This is interesting observation. Any explanations? Table 1 and 2 are interpreted from Figure 1. It cannot be stay in the main figure. If authors want to keep it, please move it to the supplementary file. The structure of the “Introduction”. I would suggest to move the sorghum paragraph into the beginning of “Introduction” after introducing the issue of salinity.
Reviewer 2 Report
The authors showed exogenous calcium treatment can improve tolerance for salinity stress conditions in Sorghum bicolor. Dissected NaCl and Ca2+ treatment supported experimental significance. They also found an appropriate Ca2+ concentration to enhance salinity tolerance for Sorghum seedlings, which could be useful for future agricultural applications.
I left a few comments for minor revision.
For Table 1
For germination rates of Mock conditions (0mM), the rate at day 3 is lower than that at day 1. Similarly, the germination rate also decreases under 150mM NaCl conditions from day3 to day7. It’s quite confusing how the germination rate decreases. I wonder if there was any technical issue. I thought the germination rate should have increased continuously because germination never moves backward.
For Table 2
I found same issue like Table 1 in 0mM CaCl2 and NaCl conditions.
Line 182-184
15mM Ca2+ also seems to inhibit total germination under 300mM NaCl conditions (Table 4)
Line 264-266
I can’t see any significant difference of MDA contents with 200mM NaCl treatment (0mM to 35mM Ca2+). Asterisk was just missing? (Figure 4d).
Some of the graphs in Figure 5 (especially figure 5e, f) would be fixed clearer. It’s hard to distinguish the difference between some bars (e.g. 300mM NaCl and 300mM NaCl + 5mM Ca2+ in Figure 5E). The authors could make some breaks in the graph to improve the figure quality.
Supplementary figures should be organized better than now. For example, I couldn’t find the word ‘Figure Sx’ except 'Figure S1’ and ‘Figure S7’, so they should be added.
Reviewer 3 Report
In their manuscript submitted to evaluation the authors present results regarding the impact of exogenously applied calcium ions to germination of Sorghum bicolor during salt stress. The plant has been very well chosen. Sorghum is a valuable, perspective plant for many reasons, including C4 photosynthesis, drought tolerance, and providing high yields of dry mass at milk-wax phase of seeds. Sorghum biomass contains high amounts of monosaccharides, what indicates its high usability for obtaining bioethanol. It is one of the most important and popular cereals in the world. It grows well in warm and hot climates so it can the basis of the diet of the inhabitants of Africa. Furthermore, as a gluten-free cereal, it is an excellent alternative to wheat. It may be as a feed plant. The experiments were planned in a proper way, they were thoroughly described and discussed. The selection of literature is correct, although several articles from the last five years are missing. Authors prose that application of exogenous calcium result in increase of salt tolerance in Sorghum what is interesting from agronomic point of view.